# Viral Load and Patterns of SARS-CoV-2 Dissemination to the Lungs, Mediastinal Lymph Nodes, and Spleen of Patients with COVID-19 Associated Lymphopenia

**DOI:** 10.3390/v13071410

**Published:** 2021-07-20

**Authors:** Adhamjon Abdullaev, Akmaljon Odilov, Maxim Ershler, Alexey Volkov, Tatiana Lipina, Tatiana Gasanova, Yuri Lebedin, Igor Babichenko, Andrey Sudarikov

**Affiliations:** 1National Research Center for Hematology, Laboratory of Molecular Hematology, Novy Zykovski Lane 4a, 125167 Moscow, Russia; adham_abdullaev@mail.ru; 2Department of Pathological Anatomy, Peoples’ Friendship University of Russia (RUDN University), 6 Miklukho-Maklaya St, 117198 Moscow, Russia; a.odilov.tma@gmail.com (A.O.); alex.volkoff@gmail.com (A.V.); babichenko@list.ru (I.B.); 3National Research Center for Hematology, Hematopoiesis Physiology Laboratory, Novy Zykovski Lane 4a, 125167 Moscow, Russia; ershler@yandex.ru; 4Department of Pathological Anatomy, Municipal Clinical Hospital Named after E.O. Mukhin, 17 Federativny Prospect, 111399 Moscow, Russia; 5Department of Cell Biology and Histology, Faculty of Biology, Lomonosov Moscow State University, Leninskie Gori, 1, 12, 119234 Moscow, Russia; tlipina@mail.ru; 6Department of Virology, Lomonosov Moscow State University, Leninskie Gori, 1, 40, 119234 Moscow, Russia; tv.gasanova@gmail.com; 7XEMA Company Limited, 9th Parkovaya St., 48, 105043 Moscow, Russia; lebedin@xema-medica.com

**Keywords:** COVID-19, SARS-CoV-2, multiplex real-time polymerase chain reaction, viral load, lymphopenia

## Abstract

Lymphopenia is a frequent hematological manifestation, associated with a severe course of COVID-19, with an insufficiently understood pathogenesis. We present molecular genetic immunohistochemical, and electron microscopic data on SARS-CoV-2 dissemination and viral load (VL) in lungs, mediastinum lymph nodes, and the spleen of 36 patients who died from COVID-19. Lymphopenia <1 × 10^9^/L was observed in 23 of 36 (63.8%) patients. In 12 of 36 cases (33%) SARS-CoV-2 was found in lung tissues only with a median VL of 239 copies (range 18–1952) SARS-CoV-2 cDNA per 100 copies of ABL1. Histomorphological changes corresponding to bronchopneumonia and the proliferative phase of DAD were observed in these cases. SARS-CoV-2 dissemination into the lungs, lymph nodes, and spleen was detected in 23 of 36 patients (58.4%) and was associated with the exudative phase of DAD in most of these cases. The median VL in the lungs was 12,116 copies (range 810–250281), lymph nodes—832 copies (range 96–11586), and spleen—71.5 copies (range 0–2899). SARS-CoV-2 in all cases belonged to the 19A strain. A immunohistochemical study revealed SARS-CoV-2 proteins in pneumocytes, alveolar macrophages, and bronchiolar epithelial cells in lung tissue, sinus histiocytes of lymph nodes, as well as cells of the Billroth pulp cords and spleen capsule. SARS-CoV-2 particles were detected by transmission electron microscopy in the cytoplasm of the endothelial cell, macrophages, and lymphocytes. The infection of lymphocytes with SARS-CoV-2 that we discovered for the first time may indicate a possible link between lymphopenia and SARS-CoV-2-mediated cytotoxic effect.

## 1. Introduction

More than 4 million patients worldwide have already died due to the COVID-19 pandemic [1]. In most patients, the leading cause of fatality was a combination of severe acute respiratory distress syndrome (ARDS) with hypercoagulation and cytopenia [2,3,4]. Lymphopenia was observed in 80% of patients with mild and 96% severe COVID-19 [5,6,7]. According to a meta-analysis of 3099 patients’ data, it was shown that lymphopenia with a level below 1000/μL was associated with a severe course of COVID-19 [8].

The pathogenesis of lymphopenia in COVID-19 may associate with more than 10 putative mechanisms [9,10,11]. One of the possible mechanisms of lymphopenia in patients with COVID-19 is extrapulmonary dissemination and direct exposure of SARS-CoV-2 particles to the tissues of secondary lymphoid organs, such as spleen and mediastinal lymph nodes (LN) filtering the lymph from the lungs [11,12].

The possibility of coronavirus dissemination to the lymph nodes was confirmed by real-time polymerase chain reaction (RT-PCR) data [12,13,14,15,16,17,18,19]. However, the studies mentioned either lack quantitation or were not normalized to the quality of the sample preparation. The involvement of the mediastinal lymph nodes in the pathological process could be indirectly evidenced by computed tomography (CT) studies since mediastinal lymphadenopathy was detected in 6% of all hospitalized patients [20,21,22] and 66% of patients with severe COVID-19 [23]. Intrathoracic lymph node enlargement was also detected during the autopsy of patients who died from COVID-19 [17,18]. Histopathological changes in the LN were characterized by pronounced edema, capillary congestion, hemorrhage [11,24], and less often by anthracosis [25]. The tissue architecture of the LN was corrupted by an indistinct division into follicles and paracortical zones [11], hypoplasia, or complete absence of germinal zones of lymphoid follicles with a decrease in the number of follicular dendritic cells and T-helpers [24]. In addition, there was a significant decrease in the number of lymphocytes with multiple foci of necrosis and apoptosis [11], pronounced sinus histiocytosis, and hemophagocytosis [11,24,25]. Histopathological changes in the spleens of patients with COVID-19 are manifested by a decrease in T and B lymphocytes, atrophy (sometimes with complete loss) of lymphoid follicles and white pulp, capillary congestion, hemorrhages, and infiltration of neutrophils, and plasma cells [11,26,27,28]. Immunohistochemical (IHC) studies of lymph nodes showing SARS-CoV-2 antigens exclusively in sinus histiocytes have been published [11,16,24,25]. IHC studies of spleen tissues revealed SARS-CoV-2 antigens in the cells of the red and white pulp of the spleen [12]. Moreover, SARS-CoV-2 RNA in the cells of the red pulp of the spleen was detected by RNA in situ hybridization [11].

Lymphopenia development due to direct infection and subsequent massive apoptosis of T-lymphocytes was previously described in related infections (SARS-CoV and MERS-CoV) [29,30,31]. Our assumptions about the possible connection of lymphopenia with the SARS-CoV-2 cytotoxic effect in infected lymphocytes are indirectly supported by the literature data [32]. There is also a report on a possible connection with the development of lymphopenia due to the SARS-CoV-2 mediated apoptosis through the p53 signaling pathway [33]. Xiang et al showed that a decrease in the number of lymphocytes could be not only a consequence of the production of IL-6 and IL-1b by SARS-CoV-2-infected macrophages and dendritic cells, but also Fas/FasL-dependent apoptosis of lymphocytes, induced by the SARS-CoV-2 [11]. Correlation of the intensity of lymphopenia with an increase of lymphocyte apoptosis in patients with severe COVID-19 was previously shown on the Iranian cohort of patients by flow cytometry [34]. However, no studies proving direct infection of lymphocytes with SARS-CoV-2 particles in patients with COVID-19 have been published so far.

The objective of the study was to assess viral load and patterns of SARS-Cov-2 dissemination to the lungs, mediastinal lymph nodes, and the spleens of patients with COVID-19 associated lymphopenia.

## 2. Materials and Methods

We used formalin-fixed and paraffin-embedded (FFPE) blocks of lung, and lymph node tissues, taken for histological studies during the autopsy of 36 patients who died of COVID-19. The study was approved by the local Medical Ethical Committee (protocol #22/2020) and carried out following the Declaration of Helsinki’s rules of 2013. According to the ninth version of the Russian Interim Guidelines for the Prevention, Diagnosis, and Treatment of New Coronavirus Infection (COVID-19) dated 26 October 2020, only tissue samples fixed in a 10% neutral buffered formalin solution for at least 24 h can be considered biologically safe.

### 2.1. RNA Extraction and cDNA Production

RNA was isolated from 10–12 3-μm-thick FFPE tissue sections of lungs, tracheobronchial lymph nodes, and spleen tissue blocks using a kit of reagents for RNA isolation from FFPE tissues, Pure Link^TM^ FFPE (Invitrogen Corporation, Carlsbad, CA, USA), according to the manufacturer’s instructions. Depending on the size of the sediment, the RNA was diluted in 15–20 µL of RNase-free water. To obtain cDNA in a volume of 20 µL, a reverse transcription reaction was performed using a 10 µL RNA solution and a set of reagents “Reverta-L” (InterLabService Ltd., Moscow, Russia), in accordance with the manufacturer’s instructions.

### 2.2. Preparation of Calibration Standards

To prepare calibration standards, cDNA fragments of SARS-CoV-2 and human *ABL1* mRNA were PCR amplified. SARS-CoV-2 406 bp long fragment (NCBI Reference Sequence: NC_045512.2 from position 3044 to 3449) was amplified with: Fw 5′-CCAGATGAGGATGAAGAAGAAGGT -3 ‘and Rv 5′-TGGCTGCATTAACAACCACTG -3′ primers and Rv 5′-CAACTGGTGTAAGTTCCATCTCT-3′ and *ABL1* (NCBI Reference Sequence: NM_005157.6) 301 bp long fragment—with Fw 5′-CGTGAGAGTGAGAGCAGTCC-3′and Rv 5′-CTGGATAATGGAGCGTGGT-3′ primers. Amplicons were cloned into pGEM^®^-T Easy plasmid vector (Promega Corporation, Madison, WI, USA) transfected and propagated, according to the manufacturer’s instructions. Serial dilutions of 100,000, 10,000, 1000, 100, and 50 copies in 10 μL for SARS-CoV-2 plasmid and 60,000, 6000, 600, and 60 copies in 10 μL for ABL1 plasmid were used for calibration as shown in Figure 1.

### 2.3. Quantitative RT-PCR

The quantitative assessment of the SARS-CoV-2 VL in tissues was carried out by the real-time quantitative polymerase chain reaction (RT qPCR) using a kit of reagents for RT-PCR (SYNTOL Ltd., Moscow, Russia) and oligonucleotide primers, as well as fluorescent probes for the SARS-CoV-2 *orf1ab* and human *ABL1* gene of the following nucleotide sequence, respectively: F1 5′-ATGATAGTCAACAAACTGTTGGTCA-3′, R1 5′-CAACTGGTTAAGTTCCATCTCT -3′, Pr1 ROX 5′-GGCAGTGAGGACAATCAGACAAC-3′BQH1 (NCBI Reference Sequence: NC_045512); F2 5′-CGTGAGAGTGAGAGCAGTCC-3′, R2 5′-GCTGGATAATGGAGCGTGGT-3′, Pr2 R6G 5′-AGCCGCTTCAACACCCTGGC-3′ BQH1 (NCBI Reference Sequence: NM_005157.6). In addition, the VL of the SARS-CoV-2 coronavirus in tissues of various organs was quantified relative to 100 copies of the ABL1 control gene using the following formula:N^SARS-CoV-2^/N^ABL1^ × 100(1)
where N^SARS-CoV-2^ is a number of SARS-CoV-2 cDNA copies, N^ABL1^ is a number of *ABL1* cDNA copies. The VL is expressed in the number of copies of SARS-CoV-2 cDNA per 100 copies of *ABL1* cDNA.

### 2.4. Sanger Sequencing of the Spike Gene

To check for possible mutations of the receptor-binding domain (RBD) and to identify SARS-CoV-2 subtype, bidirectional Sanger sequencing was performed on the overlapping Spike gene fragments, PCR amplified using the following oligonucleotide primers: 5′-CAACAGAGTTGTTATTTCTAGTGATGT-3′ and 5′-TCTGAACTCACTTTCCATCCA-3′; 5′-TGATCCATTTTTGGGTGTTT-3′ and 5′-CAGTGAAGGATTTCAACGTACA-3′; 5′-TGTGCACTTGACCCTCTCTC-3′ and 5′-TCAAAAGGTTTGAGATTAGACTTCC-3′; 5′-TCTTGATTCTAAGGTTGGTGGT-3′ and 5′-AACAGGGACTTCTGTGCAGTT-3′; 5′-TCTTTTGGTGGTGTCAGTGTT-3′ and 5′-CTGCCATATTGCAACAAAAGA-3′; 5′-GTACATTTGTGGTGATTCAACTG-3′ and 5′-CATTAAACCTATAAGCCATTTGC-3′; 5′-GCAGGTGCTGCATTACAAAT-3′ and 5′-ACCATGAGGTGCTGACTGAG-3′; 5′-CAAAAAGAGTTGATTTTTGTGGA-3′ and 5′-ATTGAGGCGGTCAATTTCTT-3′; 5′-ATGATCCTTTGCAACCTGAA-3′ and 5′-TGAAGATTCTCATAAACAAATCCA-3′.

Identification of SARS-CoV-2 subtype was carried out by comparing sequence data with the reference genome on the Nextclade v1.1.0 (https://clades.nextstrain.org/, accessed on 20 April 2021) platform.

### 2.5. Immunohistochemistry

An immunohistochemical study was performed on 3-μm-thick FFPE tissue sections onto positively charged glass slides. The detection of the coronaviral antigens was carried out using primary monoclonal antibodies against the nucleocapsid (NC) protein of SARS-CoV-2 (clone X155) produced by Xema Co. Ltd. (Moscow, Russia) in concentration of 2 μg/mL, by amplifying the signal through a multimeric detection system BenchMark ULTRA (Ventana Medical Systems, Inc., Innovation Park Drive Tucson, AZ, USA).

### 2.6. Transmission Electron Microscopy

For electron microscopy studies, lung, lymph node, and spleen tissue samples were first fixed in 4% glutaraldehyde in a 0.1 M phosphate buffer (pH 7.2–7.4) at 4 °C, and then followed postfixation in 1% osmium tetroxide at 4 °C for 2–3 h. Once the specimen was processed in a graded series of ethanol, the tissue samples were exposed to uranyl acetate for a night in 70% ethanol, dehydrated in acetone, and embedded in Epon. Epon-embedded tissue blocks were trimmed into ultrathin sections, which were then transferred to grids. The grids with ultrathin sections underwent additional contrasting in uranyl acetate and alkaline lead citrate (Reynolds, 1963). Sections were analyzed with the JEOL JEM-1400 transmission electron microscope.

### 2.7. Statistical Analysis

For statistical analysis we used free and open statistical platform jamovi (version 1.6, https://www.jamovi.org, accessed on 10 July 2021).

## 3. Results

General characteristics of the 36 patients (21 men, 15 women) with a clinical diagnosis of COVID-19, the number of bed days spent in an intensive care unit (ICU) until fatal outcome, the average blood lymphocyte counts during their stay at the ICU, the mean value of SARS-COV-2 VL in the lungs, tracheobronchial lymph nodes, and spleen are presented in Table 1.

As demonstrated in Table 1, all patients were divided into three groups according to a detection and dissemination level of SARS-CoV-2 coronavirus in tissue samples. Group 1 included three patients; no SARS-CoV-2 was detected in any organ. Group 2 consisted of 12 patients with SARS-CoV-2 dissemination limited to the lungs. Group 3 (21 patients) consisted of patients with SARS-CoV-2 disseminated to the lungs, lymph nodes, and spleen. When comparing patients from groups 2 and 3, the medians of bed days spent in ICU before death were 18.5 and 7 days, respectively. The incidences of lymphopenia <1 × 10^9^) in groups 2 and 3 were 58,3% (7/12) and 71.4% (15/21), and VLs in lungs were 18-1952 and 810-2657 SARS-CoV-2 cDNA copies per 100 *ABL1*, respectively. Moreover, we did not find any relationship of the severity of lymphopenia with the SARS-CoV-2 VL level in the lymph nodes (Pearson’s *r* = 0.117, *p*-value = 0.510) and in the spleen tissues (Pearson’s *r* = −0.176, *p*-value = 0.319).

There was a significant difference in the histological stage of inflammation in the lungs between groups 2 and 3. Dissemination of SARS-CoV-2 from the lungs into the tissue of the lymphoid organs occurs mainly during the exudative phase of DAD, when there is an active replication of SARS-CoV-2, manifested with a high level of VL SARS-CoV-2.

An example of the VL calculation for Patient 26 is illustrated in Figure 2.

The VL in Patient 26, in regard to the tissue samples of the lung, were 23,325, 59,208, 2462, 24,400 (mean VL was 27,349), lymph node was 1629, and spleen was 213 SARS-CoV-2 cDNA copies per 100 *ABL1* copies.

Sequencing data analysis has shown that SARS-COV-2 in all patients was of the 19A subtype. Patient 13 was co-infected with two viruses, with mutations c.23191C and c.23191T, respectively (Figure 3).

To confirm qPCR results IHC study was conducted using monoclonal antibodies against the NC protein of coronavirus SARS-CoV-2. IHC staining of lungs, lymph nodes, and the spleen of the patient with a high VL is shown in Figure 4.

IHC study with mAb against SARS-CoV-2 NC protein in lung tissue showed diffuse positive staining of the cytoplasm of type 1 pneumocytes with a flattened nucleus and a thin elongated cytoplasm, and larger and thickened type 2 pneumocytes (Figure 4A,B), alveolar macrophages, as well as atypical multinuclear cells (Figure 4E). In areas of lung tissue with an enlarged and thickened alveolar septum, there was a bright positive reaction in the part of interstitial cells and pneumocytes (Figure 4C). Immunoreactivity to anti-NC mAb was also found in single cells of the partially desquamated prismatic epithelium of bronchioles (Figure 4D). IHC examination of tracheobronchial lymph nodes revealed SARS-CoV-2 antigens only in sinus histiocytes (Figure 4F,G). In the spleen, positive staining was observed mainly in the cells of Billroth’s cords of the red pulp and single cells of the spleen capsule (Figure 4H,I). No immunoreactivity to anti-NC mAb was detected in the white pulp (Figure 4J).

To confirm the above results, we decided to perform direct visualization of SARS-CoV-2 viral particles by transmission electron microscopy (TEM). The results for the tissues of the lungs and mediastinal lymph nodes are shown in Figure 5.

TEM examination of lung tissue revealed an endothelial cell with a vesicle in the cytoplasm containing coronavirus particles characterized by a spherical shape and an average size of 100 nm. The particles were surrounded by a membrane with electron-dense outgrowths of the S-protein and granular structures of the nucleocapsid visualized in the lumen of the particles. SARS-CoV-2 virions have also been found in the cytoplasm of lymph node macrophages within perinuclear multivesicular structures associated with the granular endoplasmic reticulum and the Golgi complex. In addition, a vesicle with multiple particles of the SARS-CoV-2 coronavirus was found in the cytoplasm of the lymphocyte. The vesicle with virions was located perinuclear and was associated with the granular endoplasmic reticulum.

## 4. Discussion

According to several studies, lymphopenia occurs in 80% of patients with mild and 96% severe COVID-19 and is a prognostic marker of poor disease outcome [5,6,7,35]. However, the pathogenesis of lymphopenia in COVID-19 could have complex etiology and may include more than 10 putative mechanisms. In particular, the development of lymphopenia in patients with COVID-19 could result from extrapulmonary dissemination and the direct effect of SARS-CoV-2 particles on the tissues of the mediastinal lymph nodes, which filter lymph flowing from the lung primary inflammatory focus, and spleen, as a significant human lymphoid organ [9,10,11,35].

Possible dissemination of coronavirus in lymph nodes tissue based on RT-PCR data was previously reported [12,13,14,15,16]. However, Ct threshold measuring applied in these studies might have limited probative value since underestimation of low template counts [12,13,14,15,16,17,18,19]. Moreover, according to some authors, the values of the threshold cycle of RT-PCR should be interpreted critically since Ct is strongly influenced by numerous factors [36,37]. Here, we present an original RT-PCR method of SARS-CoV-2 quantitation based on the measurement of true copies of SARS-CoV-2 cDNA relative to copies of *ABL1* cDNA. For assay calibration, plasmids with cDNA inserts of SARS-CoV-2 and *ABL1* gene fragments were constructed.

The incidence of lymphopenia (<1 × 10^9^/L) in our cohort of patients with COVID-19 was 63.8% (23/36), which is much higher than that noted in other studies [35,38]. The incidence of viral dissemination to lungs, tracheobronchial lymph nodes, and spleen was 91.2% (33/36), 55.8% (19/34), and 35.3% (12/34), respectively. Despite the assertion of some authors that, lymphopenia in COVID-19 is due to the direct damaging effect of SARS-CoV-2 to the tissue of secondary lymphoid organs [11,12], in our study lymphopenia <1 × 109/L was observed both in patients with dissemination to lymphoid organs and without. Particularly in the group of patients without dissemination of SARS-CoV-2 in the LN and spleen, lymphopenia was observed in 7 of 12 patients (58.3%). Moreover, we did not find any relationship of the level of SARS-CoV-2 VL in lymphatic tissue with the severity of lymphopenia. Conversely, Patients 3, 24, and 35, with the dissemination of SARS-CoV-2 in the LN and spleen tissue, did not develop lymphopenia <1 × 10^9^/L, while in Patients 6, 11, 20, and 29, without viral dissemination, lymphopenia <1 × 10^9^/Lwas observed.

The absence of extrapulmonary dissemination was detected in 33.3% (12/36) of patients (second group), and the median SARS-CoV-2 VL was 239 (range 18–1952) copies of SARS-CoV-2 cDNA per 100 copies of *ABL1*. At the same time, the histomorphological changes in the lungs corresponded to bronchopneumonia and the proliferative phase of DAD. Dissemination of SARS-CoV-2 to the lung tissue, mediastinal LN, and spleen tissues was detected in 58.4% (21/36) patients (third group). The median VL in the lungs was 12,116 copies (range 810–250281), lymph nodes–832 copies (range 96–11586), and spleen–71.5 copies (range 0–2899). Thus, the histological changes in the lungs in most of the studied samples of this group of patients corresponded to the exudative phase of DAD.

The medians of bed days spent in ICU before the death of the second and third group of patients were 18.5 and 7 days, respectively. A shorter period of stay in ICU was associated with higher SARS-CoV-2 VL in the lungs and extrapulmonary dissemination of the virus to secondary lymphoid organs.

In the revealed case of coinfection with SARS-CoV-2 c.23191C and c.23191T variants, the amino acid structures of the Spike proteins were identical since the SARS-CoV-2 c.23191C/T mutation is synonymous. We exclude PCR cross-contamination, as this is the only case of the c.23191T variant detected in our laboratory. This case can be explained by the fact that the SARS-CoV-2 coronavirus is characterized by a high frequency of mutations with the formation of quasispecies (subpopulations) in the body of the host [39,40,41].

IHC study of the lung tissue with anti-NC mAb revealed immunoreactivity in the cytoplasm of type 1 and 2 pneumocytes, alveolar macrophages, and bronchiolar epithelium cells, which, according to other studies, is determined in 73-85.7% of cases [12,16,22,23]. The SARS-CoV-2-positive atypical multinuclear cells that we discovered are of epithelial origin [16] and are formed due to the fusion of adjacent cells infected with SARS-CoV-2 [42]. In the tissue of the lymph nodes, we, as well as other authors [11,16,24,25], managed to identify the SARS-CoV-2 antigens in the cytoplasm of exclusively sinus histiocytes. In spleen tissue, positive staining with anti-NC SARS-CoV-2 mAb was observed in cells of predominantly red pulp and single cells of the spleen capsule but was not detected in the white pulp. We could not find in the available literature data on the IHC study of the spleens of patients with COVID-19, except for the only unpublished article by Feng and co-authors [12]. They described immunoreactivity to anti-SARS-CoV-2 antibodies in both red and white pulp.

The ultrastructural morphology of coronavirus particles we obtained by TEM examination of the tissue of the lungs and lymph nodes is similar to the data reported by other researchers, both in native tissues and in cell cultures [43,44,45]. The detection of SARS-CoV-2 virions in the cytoplasm of endothelial cells and its possible impact on the development of systemic pathology have been previously described by other authors [46,47,48,49]. The mechanism of lymphopenia due to the direct viral infection of T-lymphocytes and subsequent massive apoptosis has been described as related to the SARS-COV-2 coronaviruses MERS-CoV and SARS-CoV [29,30,31]. Recently an article describing SARS-CoV-2 mediated apoptosis through the p53 signaling pathway as a potential cause of lymphopenia has been published by Xiang et al. [33] A decrease in the number of lymphocytes shown to be caused by the production of IL-6 and IL-1b by SARS-CoV-2-infected macrophages and dendritic cells and, more important, by Fas/FasL-dependent apoptosis of lymphocytes, induced by the SARS-CoV-2 [11]. Previously it was shown by flow cytometry that the severity of lymphopenia is correlated with lymphocyte apoptosis in Iranian patients with severe COVID-19 [34]. However, the vesicle with multiple particles of coronavirus identified by us in the lymphocyte cytoplasm, as far as we know from the available literature, is the first report of direct lymphocyte infection with SARS-COV-2 coronavirus in patients with COVID-19. Thus, for the first time, we confirmed the possibility of direct infection and active replication of the SARS-CoV-2 coronavirus in lymphocytes in patients with COVID-19.

## 5. Conclusions

The proposed SARS-COV-2 quantitation method based on serial dilutions of cloned calibrator templates (cDNA human *ABL1* gene cDNA) is suitable for the relative SARS-COV-2 copy number estimation. The development and severity of lymphopenia <1 × 10^9^/L in patients with fatal COVID-19 outcomes are not associated with the pattern of dissemination and the level of VL SARS-CoV-2 in the tissues of the mediastinal LN and spleen. Extrapulmonary dissemination of SARS-CoV-2 in the tissue of the lymphatic organs occurs during the exudative phase of DAD. It is associated with a shorter period of stay in ICU and higher SARS-CoV-2 VL in the lung tissues compared with patients with SARS-CoV-2 dissemination limited to the lungs. The finding of a lymphocyte infected with SARS-COV-2 particles, which we first revealed by TEM, might indicate that SARS-COV-2 could induce lymphopenia through possible direct cytotoxic effect.

## Figures and Tables

**Figure 1 viruses-13-01410-f001:**
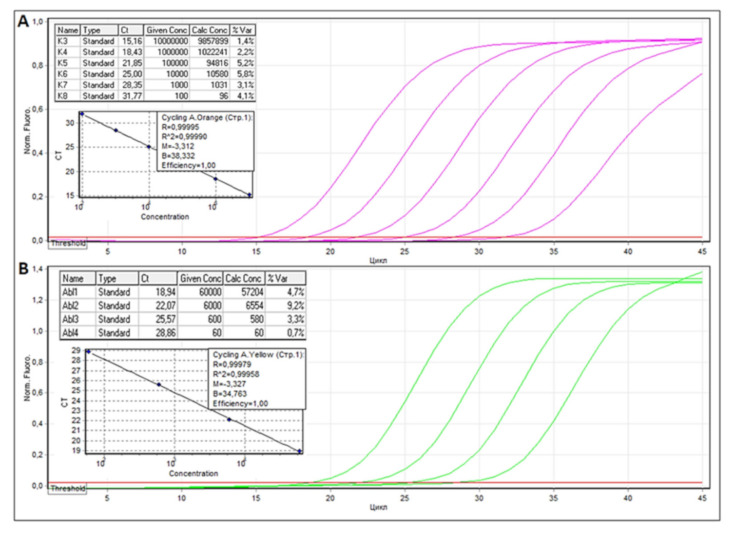
Threshold cycles and graphical RT qPCR curves for SARS-CoV-2 (**A**) and *ABL1* (**B**) calibrator plasmids.

**Figure 2 viruses-13-01410-f002:**
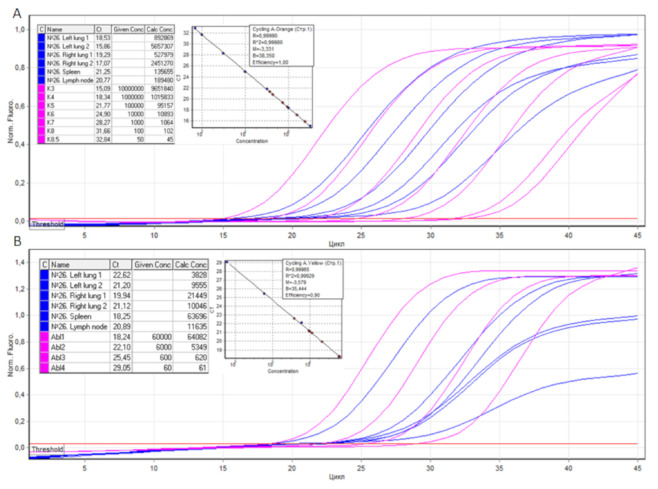
PCR threshold cycles for SARS-CoV-2 and *ABL1* cDNAs isolated from the lungs, spleen, and lymph nodes of Patient 26. (**A**) The number of SARS-CoV-2 cDNA copies (with R^2^ = 0.99980 and an efficiency of 1.00) of the left and right lung tissue samples were 892,869, 5,657,307, and 527,979, 2,451,270, respectively. The number of SARS-CoV-2 cDNA copies of the spleen and lymph node tissue samples was 135,655 and 189,480, respectively. (**B**) The number of *ABL1* cDNA copies (with R^2^ = 0.99929 and an efficiency of 0.90) of the left and right lung tissue samples were 3828, 9555, and 21,449, 10,046, respectively. The number of *ABL1* cDNA copies of the spleen and lymph node tissue samples was 63,696 and 11,635, respectively.

**Figure 3 viruses-13-01410-f003:**
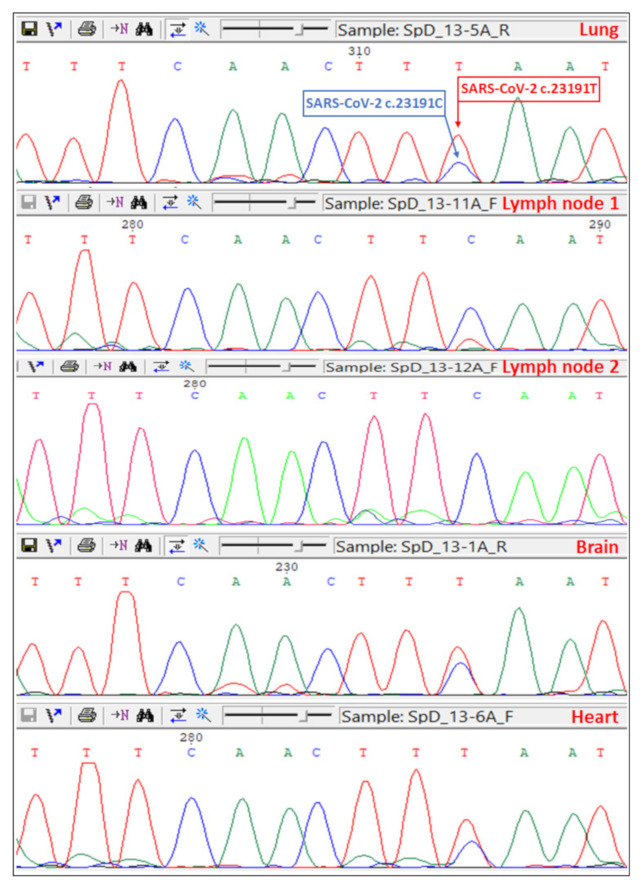
Spike gene Sanger sequencing chromatograms of lungs, lymph nodes, brain, and heart tissues of Patient 13.

**Figure 4 viruses-13-01410-f004:**
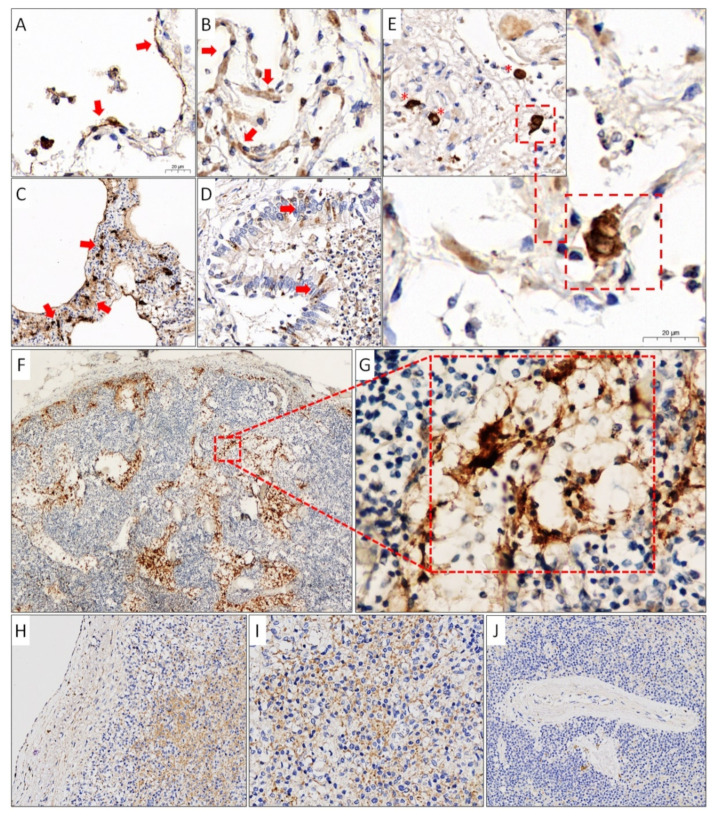
Results of an immunohistochemical study of lung tissues (**A–E**), tracheobronchial lymph nodes (**F**,**G**), and spleen (**H–J**) using mAb against SARS-CoV-2 NC protein. (**A**) Positive staining of type 1 pneumocytes (red arrows) and cells in the lumen of the alveoli (×600). (**B**) Positive staining of type 2 pneumocytes (red arrows) (×600). (**C**) Positive staining of cells of the dilated and thickened alveolar septum (red arrows). (**D**) Desquamated bronchiolar prismatic epithelium with single positively stained cells (red arrows) (×400). (**E**) Positive staining for alveolar macrophages (red asterisks) and multinuclear cells (red frame) (×400). (**F**,**G**) Positive staining of hyperplastic sinus histiocytes in the lumen of dilated sinuses of the lymph node (×40, ×400). (**H**,**I**) Positive staining of cells of the Billroth pulp cords, as well as single cells of the spleen capsule (×200, ×400). (**J**) Negative staining of cells of the periarterial lymphoid sheaths (×200).

**Figure 5 viruses-13-01410-f005:**
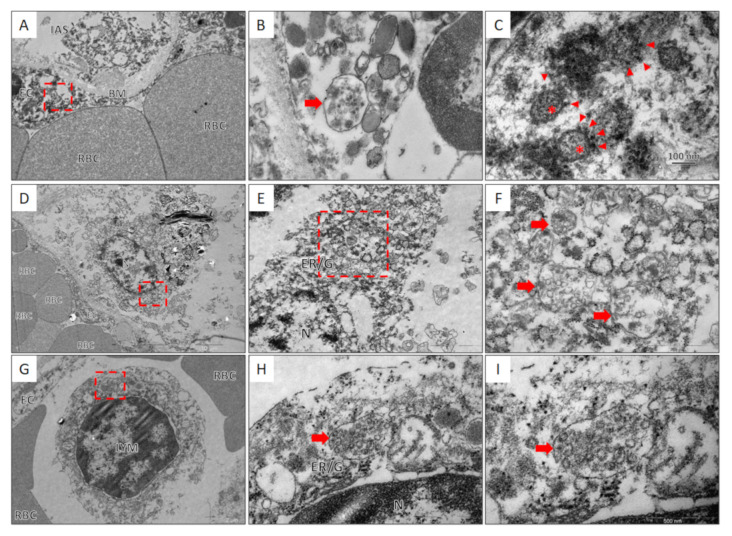
Results of TEM examination of lung tissue (**A–C**), tracheobronchial lymph nodes (**D–F**), and lymphocyte infected with SARS-CoV-2 coronavirus (**G–I**). (**A**,**B**) The section of the air-blood barrier is divided by the basement membrane (BM) into the intra-alveolar space (IAS) and the lumen of the capillary with red blood cells (RBC). In the cytoplasm of the endothelial cell (EC) (red frame), a membrane vesicle with spherical coronavirus particles inside is determined (red arrow). (**C**) At high magnification, the coronavirus particles were spherical, with an average size of 100 nm, surrounded by a membrane, on the surface of which there are electron-dense outgrowths of the S-protein (arrowhead), and granular nucleocapsid structures (asterisks) can be seen in the lumen of the particles. (**D**,**E**) In the lymph node, in the cytoplasm of a pericapillary macrophage (small red frame), multivesicular structures (large red frame) associated with the endoplasmic reticulum and cisterns of the Golgi complex (ER/G) were identified. (F) At high magnification, SARS-CoV-2 virions (red arrows) were seen inside the vesicles. (**G–I**) In the lumen of the capillary, along with erythrocytes (RBC), a lymphocyte (LYM) was found, in the cytoplasm of which (red frame) perinuclearly bound, a vesicle (red arrow) containing coronavirus particles was found.

**Table 1 viruses-13-01410-t001:** General characteristics of patients and SARS-CoV-2 VL in lung, lymph node, and spleen tissues.

#	Sex	Age	Time ^1^	LYM Count ^2^	Phase of DAD ^3^/Other Pulmonary Findings	*X*^VL^ in the Lungs ^4^	VL in LN ^5^	VL in Spleen ^6^
	*Group 1: (n = 3) SARS-CoV-2 has not been detected*							
7	М	62	2	0.9	Proliferative	0	0	0
17	F	77	15	1.3	Proliferative/emphysema	0	0	0
23	М	90	9	2.4	Bronchopneumonia with hemorrhages and fibrosis	0	0	0
	*Group 2: (n = 12) SARS-CoV-2 dissemination limited to lungs*							
5	F	73	18	4	Proliferative/bronchopneumonia with hemorrhages	18	0	0
34	F	73	23	1.3	Bronchopneumonia	27	0	0
29	F	79	24	0.7	Bronchopneumonia	38	0	0
22	М	90	10	1.9	Proliferative	63	0	0
15	М	68	19	0.9	Bronchopneumonia	190	0	0
31	F	78	10	0.8	Proliferative	208	0	0
20	F	85	23	0.4	Bronchopneumonia	270	0	0
6	М	66	27	0.3	Hemorrhages and fibrosis	313	0	0
8	М	64	19	1.2	Proliferative	677	0	0
32	М	85	10	1.6	Proliferative	694	0	0
11	М	84	17	0.7	Proliferative	706	-	n/a
14	М	80	10	0.6	Proliferative	1952	0	0
*Group 3: (n = 21) SARS-CoV-2 dissemination to lungs, lymph nodes, and spleen*								
33	F	86	3	0.6	Proliferative	810	96	0
13	F	71	1	0.7	Exudative	834	1318	629
35	F	73	2	1.4	Proliferative	910	215	197
21	М	67	13	0.4	Proliferative	982	180	0
30	М	61	5	1.3	Proliferative/bronchopneumonia	1002	237	0
1	М	52	1	2.7	Proliferative/emphysema	1016	112	0
25	М	74	22	0.5	Proliferative	1032	825	0
16	М	67	9	0.7	Proliferative/bronchopneumonia with hemorrhages	2657	1110	0
4	М	76	5	0.8	Exudative and early proliferative/hemorrhages	4909	195	71
19	М	84	24	0.5	Exudative	7476	1807	0
9	F	72	12	0.6	Exudative/bronchopneumonia	12,116	3801	-
28	F	95	7	0.4	Exudative	14,522	839	34
3	М	84	11	1.2	Exudative and early proliferative/bronchopneumonia with necrosis	14,937	162	9
27	М	84	16	0.3	Exudative	16,867	1180	1320
18	М	85	16	0.5	Exudative	17,817	n/a	218
36	М	73	2	0.6	Exudative	18,219	2691	181
26	F	93	7	0.9	Exudative	27,349	1629	213
2	F	86	25	1	Exudative	73,214	789	72
12	F	85	7	0.9	Exudative	151,183	513	1891
24	F	76	11	2.7	Exudative	159,217	11,586	551
10	М	61	6	0.4	Exudative	250,281	5958	2899

^1^ Number of days spent in ICU before the fatal outcome; ^2^ Lymphocytes count (10^9^/L of blood); ^3^ DAD, diffuse alveolar damage; ^4^ Mean VL level in lung samples; ^5,6^ VL level in the lymph nodes and spleen; “n/a”—material was not available; “-”—no PCR amplification.

## Data Availability

The data presented in this study are available on request from the corresponding author.

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
