# Peer review of "Viral Load and Patterns of SARS-CoV-2 Dissemination to the Lungs, Mediastinal Lymph Nodes, and Spleen of Patients with COVID-19 Associated Lymphopenia"

_viruses, 2021, doi:10.3390/v13071410_

Round 1

Reviewer 1 Report

In this manuscript titled “Viral load and patterns of SARS-CoV-2 dissemination to the 2 lungs, mediastinal lymph nodes, and spleen of patients with 3 COVID-19 associated lymphopenia”, the authors analyzed the association between viral load of SARS-CoV-2 in different organs and COVID-19 related lymphopenia. They provided the molecular genetic, immunohistochemical and electron microscopic data to support the conclusion that SARS-CoV-2 could directly infect the lymphocytes leading to lymphopenia. The study is interesting, while several key experiments are missing to fully support the conclusion.

  1. The authors stated that there are no association between the severity of lymphopenia in patients and the level of VL SARS-CoV-2 in the tissues of the mediastinal LN and spleen. I would recommend the authors to perform the correlation analysis to support the conclusion.
  2. In section 2.2, the authors need to clarify the location of the amplified fragment in SARS-CoV-2 genome.
  3. The authors need to provide the evidence showing that ABL1 mRNA level in different organs is comparable.
  4. In Figure 5, there are no clear coronavirus-like virus particles observed, it’s better for the authors to perform the Immunogold labeling to confirm the results.

Reviewer 2 Report

Major points:

  • It is hardly difficult to read, understand, and appreciate an original manuscript on COVID-19 which has (as per the lines 86-93) four different objectives, let alone many of which have been partially addressed in the literature.
  • The finding of direct lymhotropic action of SARS-CoV-2 is quite interesting but additional markers (e.g., apoptotic markers) would have added major value to this hypothesis 
  • With the above in mind, the conclusion in line 38-41 is a quite substantial jump from the already existing literature. 
  • Overall, I would recommend significantly restructuring the whole narrative of the manuscript before reconsidering this manuscript in any publication venue. 
  • It is still unclear to me how the authors divided between patients with lympopenia and SARS-CoV-2 and why they did not concentrate their studies on these patients, since it appears that some results are mixed (i.e., from both groups). 
  • Importantly, there appears to be no sort of statistical analysis in the paper to make some comparisons.                                                          Minor points:
  • There are some linguistic errors, which need correction (e.g., line 58, it should be "was confirmed", etc. These errors are minor and have not affected my general judgment of the manuscript.
  • I would prefer if a graphical abstract/ figure/ photo/ schema on the multiple ways that "link" SARS-CoV-2 and lymphopenia was created.

Reviewer 3 Report

One of the parameters that we will probably have to use in the future for the study of patients with COVID-19 is the viral load. It may be one of the explanations for the clinical evolution of many patients, and may also help to monitor specific therapies when they become available. The authors adequately analyze the problem by focusing on lymphopenia.
The article seems to me relevant, adequately structured and with conclusions that show the usefulness of the technique.
I would accept the article without modification

Author Response

We really appreciate friendly and positive comments from a respected reviewer and high estimate of our modest work.

Comments and Suggestions for Authors

One of the parameters that we will probably have to use in the future for the study of patients with COVID-19 is the viral load. It may be one of the explanations for the clinical evolution of many patients, and may also help to monitor specific therapies when they become available. The authors adequately analyze the problem by focusing on lymphopenia.
The article seems to me relevant, adequately structured and with conclusions that show the usefulness of the technique.
I would accept the article without modification

Round 2

Reviewer 1 Report

The revised manuscript is ready for publication

Reviewer 2 Report

I am still not satisfied with this article, and I leave the final decision to the Editor.